# Beyond the transcript: Chromatin implications in trans-splicing in Trypanosomatids

**Romina Trinidad Zambrano Siri[1,2], Paula Beati[1], Lucas Inchausti[3,4], Pablo Smircich[3,4], Guillermo Daniel Alonso[1,2]\*, Josefina Ocampo[1]\***

**1** Instituto de Investigaciones en Ingeniería Genética y Biología Molecular "Dr. Héctor N. Torres" (INGEBI), Consejo Nacional de Investigaciones Científicas y Técnicas (CONICET), Buenos Aires, Argentina, **2** Departamento de Fisiología, Biología Molecular y Celular, Facultad de Ciencias Exactas y Naturales, Universidad de Buenos Aires, Buenos Aires, Argentina, **3** Laboratorio de Bioinformática, Departamento de Genómica, Instituto de Investigaciones Biológicas Clemente Estable (IIBCE), Montevideo, Uruguay, **4** Sección Genómica Funcional, Facultad de Ciencias, Universidad de la República (UdelaR), Montevideo, Uruguay

\* galonso@dna.uba.ar (GDA); jocampo@ingebi-conicet.gov.ar (JO)

## Abstract

*Trypanosoma cruzi, Trypanosoma brucei* and *Leishmania major*, usually known as TriTryps, are the causal agents of animal and human sickness, and are characterized by having complex life cycles, alternating between a mammalian host and an insect vector. Their genes are organized in long transcriptional units that give rise to polycistronic transcripts which maturate into mRNA by a process known as trans-splicing. Among those genes, an important subset is composed of multi-copy genes, which play crucial roles in host invasion and immune evasion. Here, we predicted the most likely trans-splicing acceptor sites **(TASs)** for TriTryps and found that the average chromatin organization is very similar among them with a mild nucleosome depletion at the TASs, and the same layout is observed in most of the genome. A detailed examination of the nucleosome landscapes resulting from different levels of chromatin digestion in *T. brucei* shows that an MNase-sensitive complex is protecting the TASs, and it is at least partly composed of histones. Additionally, comparative analysis for single and multi-copy genes in *T. cruzi* revealed a differential chromatin structure at the TASs suggesting a novel mechanism to guarantee the fidelity of trans-splicing in trypanosomatids.

## Introduction

The protozoan parasites *Trypanosoma cruzi*, *Trypanosoma brucei* and *Leishmania major*, usually known as TriTryps, are responsible for Chagas disease, Sleeping sickness, and Leishmaniasis respectively. They belong to the group of neglected tropical diseases and together affect more than 30 million people worldwide [1].

These parasites have complex life cycles alternating between an insect vector and a mammalian host. To cope with this alternation through different hostile

**Data availability statement:** "All relevant data are within the manuscript and its Supporting Information files or extracted from previous published articles. A source code for the analytical procedures here described is available at https://github.com/romizambrano/TAS".

**Funding:** International Development Research Centre (IDRC):Guillermo D. Alonso 109929; Agencia Nacional de Promoción de la Investigación, el DesarrolloTecnológico y la Innovación (agenciaidiar):Josefina Ocampo PICT-2020-00473; Consejo Nacional de Investigaciones Científicas y Técnicas (CONICET):Josefina Ocampo PIBBA 2020-28720210100100CO; Consejo Nacional de Investigaciones Científicas y Técnicas (CONICET):Guillermo D. Alonso PIP 2021-2023-11220200103073CO; Agencia Nacional de Promoción de la Investigación, el Desarrollo Tecnológico y la Innovación (agenciaidiar): Guillermo D. Alonso, PICT Raíces 2019-4260. The funders had no role in study design, data collection and analysis, decision to publish, or preparation of the manuscript.

**Competing interests:** The authors have declared that no competing interests exist.

environments, TriTryps need to adapt their gene expression to the specific requirements and challenges they face, including being able to achieve host invasion. Among the strategies orchestrated to achieve this aim, these parasites count on multi-copy genes encoding for specialized surface proteins involved in virulence or immune system evasion [2–6].

One peculiarity of TriTryps is that their genes are organized into directional gene clusters (DGCs) that encode polycistronic transcription units which are further processed into monocistronic units by a co-transcriptional process that involves polyadenylation and trans-splicing [7]. This maturation involves a cleavage event coupled to the addition of a Spliced Leader sequence at the 5'-end and a poly(A) tail at the 3'-end of each mRNA [8]. Although in TriTryps gene expression is mainly regulated post-transcriptionally, there is substantial evidence that chromatin punctuation by epigenetic factors exerts additional modulation [9–15].

In eukaryotic cells, chromatin arrangements are well orchestrated in the nucleus to regulate DNA exposure, so nucleosomes have been proposed as major determinants of DNA accessibility [16]. While *in vitro*, DNA sequence preferences determine nucleosome formation, *in vivo*, the outcome will be the result of the interplay with ATP-dependent chromatin remodeling complexes, non-histone DNA binding proteins, histone variants, histones post-translational modifications, and transcription [17–19]. Regarding the influence of DNA sequences, regions rich in poly AT or poly GC are refractory to bend around the histone octamer [20–23], while tracks of DNA with a 10 bp periodicity of AT dinucleotides facilitate the bending of the DNA around the histone core [24,25].

In TriTryps, it has been previously observed that dinucleotide repeats are non-uniformly distributed along the DGCs, suggesting that DNA sequence composition might influence genome compartmentalization and gene expression [26]. Additionally, it was described that the genome of *T. cruzi* is compartmentalized into a 'core compartment', with lower GC content mainly harboring conserved single-copy genes; and a 'disruptive compartment,' which exhibits high GC content and is mainly composed of multi-copy genes [27].

In general, nucleosome distribution on DNA sequences is organized into regular arrays where every nucleosome is spaced from the neighboring ones by a stretch of DNA called linker. In model organisms, nucleosomes are regularly spaced and phased over coding regions relative to the transcription start site of genes and present nucleosome depleted regions (NDRs) at promoters surrounded by well-positioned nucleosomes at +1 and −1 position [28–30]. The general bases of chromatin landscape are conserved from yeast to humans, but in more complex organisms this regular pattern is mainly associated with highly transcribed genes, while silenced genes usually do not have clear NDR or phased nucleosomes [31].

In trypanosomes, there are no canonical promoter regions. Instead, transcription is initiated from dispersed promoters and in general they coincide with divergent strand switch regions; such initiation sites also include single transcription start regions [32,33]. More recently, it was proposed that two promoters are located between divergent gene clusters driving unidirectional transcription [34]. Consistent with an ongoing

passage of RNA polymerase II, the nucleosome maps reported for TriTryps revealed poor nucleosome organization with no average spacing or phasing [33,35–37]. From our perspective, the most relevant observation made from the previous chromatin studies in TriTryps is that average nucleosome occupancy changes around the TASs, suggesting a potential role of chromatin in trans-splicing. However, a proper comparison of the nucleosome maps from the three organisms is missing.

In this work, we performed a thorough comparison of MNase-seq data publicly available for the stages present in the insect vectors in TriTryps, generated by others and in our laboratory [35–38]. To enable a meaningful and unbiased comparison across species, we developed a unified and systematic analysis pipeline and applied it consistently to all datasets. While previous studies analyzed each trypanosomatid independently, often using distinct experimental and computational methodologies, here all raw MNase-seq data were reprocessed using identical parameters and analytical steps for each TriTryp. This approach minimizes methodological variability and allows direct cross-species comparisons. Consistent with the original works, from average nucleosome occupancy plots we observed a mild NDR at the TASs in *T. cruzi*, a shallower trough preceded by an MNase protected footprint in *L. major*, and an MNase protection at the TAS in *T. brucei.* Nevertheless, when analyzing comparable levels of digested chromatin, we unveiled that both, *T. cruzi* and *T. brucei,* present a nucleosome depletion at the TASs. Additionally, we analyze different levels of MNase-digested samples for *T. brucei* and we demonstrate that an MNase-sensitive complex is protecting the TASs. This complex is at least partly composed of histones as shown by MNase-ChIP-seq data for histone H3 and is detected both in *T. brucei* and *T. cruzi*, suggesting a conserved protection of the TASs and sensitivity to MNase in trypanosomes. Moreover, comparative analysis for single and multi-copy genes in *T. cruzi* revealed that their TASs are differentially protected from MNase digestion. This observation unveils that the NDRs formation at the TASs in epimastigotes occurs more efficiently at genes that need to be expressed at that life-stage. Furthermore, by analyzing dinucleotide frequencies around TASs we observe different patterns for single and multi-copy genes, possibly implying that transcript maturation is additionally granted by the underlying DNA sequence composition in a stage-independent manner.

## Materials and methods

### Data collection

Informatics analysis were performed using publicly available data from Gene Expression Omnibus [39] as detailed in S1 Table. MNase-seq and MNase-ChIP-seq data available for the parasite stadium detected in the insect vector for *T. cruzi* CL Brener strain, *T. brucei* 427 strain and *L. major* Friedlin strain were used for chromatin studies [35,38,40–43].

### Statistical analysis, genome alignment and reference genomes

The sequence quality metrics were assessed using FastQC v0.11.9 (https://www.bioinformatics.babraham.ac.uk/projects/fastqc/). During this step, over represented sequences were detected and trimmed out using Cutadapt tool v3.5 (https://cutadapt.readthedocs.io/en/stable/) [44] when required.

Paired-end reads were aligned using Bowtie2 v2.4.4 (https://bowtie-bio.sourceforge.net/bowtie2/index.shtml) [45] against version TriTryp46 of the respective genomes retrieved from TriTrypDB [46].

For *T. cruzi* CL Brener, we built a genome combining the Esmeraldo-like haplotype, the non Esmeraldo-like haplotype and the extra regions, not assigned to any haplotype, as described before [40]. For *T. brucei* 427 the corresponding genome was used, compatible with a software applied in further steps. For *L. major* Friedlin its corresponding genome was used [47].

### TAS prediction

The most likely trans-splicing acceptor site (TAS) for each member of TriTryp was predicted as described before [40]. Briefly, the 5′untranslated regions (5′UTR) were predicted with UTRme [48]. Given the lack of transcriptomic data for

CL Brener epimastigotes, the predictions were based on available RNA-seq data for the Y strain [49] using TriTryp46 Esmeraldo-like genome as a reference. In the case of *T. brucei* predictions were based on Lister 427 strain data [13] using TriTryp46 *T. brucei* 427 genome. In the case of *L. major* Friedlin, predictions were based on available data for this strain [50] using TriTryp46 *L. major* Friedlin genome. As an approximation of the trans-splicing acceptor site, the 5' end of the 5'UTR region was used (Fig. 1A). The list of the genomic coordinates for the predictions for each TriTryp is provided in S2 Table.

### Data visualization and genomic signal profiling

Length distribution heatmaps, BigWig files, average occupancy plots and 2D-plots were generated from BAM files as described before [40,51]. Briefly, BigWig files containing information of nucleosome occupancy (MNase-seq) and histone H3 signal (from MNase-ChIP-seq) were generated by counting the number of times that a base pair was occupied. The signals were normalized by summing all the sequences covering a nucleotide and dividing that number by the average number of detected sequences per base pair across the genome. For MNase-seq and H3-IP the analysis was performed for all the sequenced fragments (50–500 bp) or restricted to those fragments that belong to the dinucleosome-range (180–300 bp), nucleosome-size range (120–180 bp), or subnucleosome-range (50–120 bp), as detailed in the figure legends.

Average occupancy plot, 2D-plots and heatmaps were built using the TASs predictions with best score, obtained for each parasite, as reference point. In the case of *T. cruzi* only the Esmeraldo-like haplotype was represented but the alignments of the fastq files were made to the whole genome as explained above.

For 2D-plots, the data was represented relative to the TAS in the x axis, while the size of the analyzed DNA fragments was represented in the y axis. A source code easy to adapt to any TriTryp is available at (https://github.com/paulati/nucleosome).

To build heatmaps for the disaggregated regions around TAS, computeMatrix and plotHeatmap functions from deepTools version 3.5.1 (https://test-argparse-readoc.readthedocs.io/en/latest/) were used [52]. BED6 files containing the genomic coordinates for the TASs predictions with best score were used as regions (detailed in S2 Table) and BigWig files generated for each analyzed dataset were used as score files.

Average plots and heatmaps were represented either for all the genes with a predicted TASs or sorted for single and multi-copy genes of *T. cruzi* when stated.

### TAS regions definition and Dinucleotide frequency analysis

The region surrounding every predicted TAS in a 100 bp window was annotated in a BED format where the start site was the position located 100 bp upstream of the TAS, and the end site was the position located 100 bp downstream of the TAS. The sequences of these regions, named **TAS regions**, were then saved in a FASTA format and used for further analysis.

The periodicity of the AA/TT/AT/TA, GG/CC/GC/CG or other possible combinations of dinucleotides in the TAS region was analyzed and the average frequency of these dinucleotides was represented. The occurrence of undetermined dinucleotides "NN", based on the genome annotations, was considered.

The procedure used for the analysis of TAS regions and dinucleotide frequencies is available at https://github.com/romizambrano/TAS.

### Construction of gene subsets

GFF files with TAS predictions for *T. cruzi* were obtained using UTRme as described above and are annotated in S2 Table. Single/low and multi-copy genes lists for *T. cruzi* were generated by filtering the TAS prediction as follow: Trans-sialidase, trans-sialidase-like, MASP, Mucins, GP63, retrotransposon hot spots or RHS, and dispersed gene family or

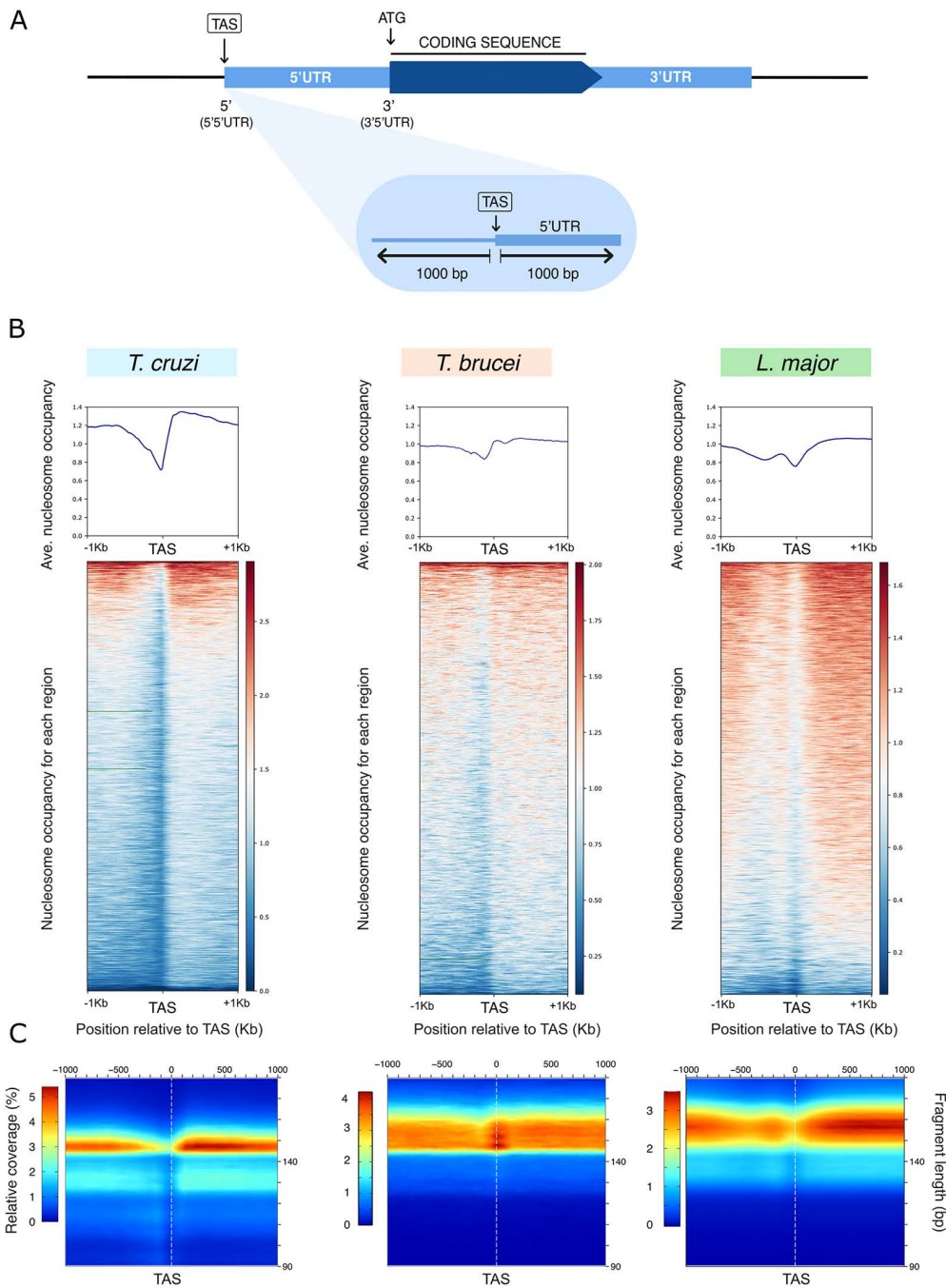

**Fig 1. Chromatin organization around TASs in each TriTryp.** (A) Schematic representation of TASs prediction from 5'UTR regions obtained with UTRme (Radio et al., 2017). (B) Average nucleosome occupancy (top panels), heatmaps for each region in a 1 kb window (bottom panels). The signals scored for DNA molecules in the nucleosomal-size range (120-180 bp) are represented. (C) 2D occupancy plots showing nucleosome density relative to the TAS (bottom panels) for one representative data set of *T. cruzi* (GSM5363006), *T. brucei* (GSM2407366) and *L. major* (GSM2179742) respectively. Red: High nucleosome density; blue: low nucleosome density.

DGF-1 were categorized as multigene families. Gene IDs corresponding to these multigene families were obtained by text searches using the current genome annotation on the "description" field in the gff file. Multi-copy genes were defined as those whose gene IDs belonged to the latter gene list. Those genes that did not meet this criterion were defined as single/low-copy genes. For simplicity, "single-copy" will be used to refer to these genes throughout the manuscript. Both single-copy and multi-copy genes are detailed in S3 Table.

For the subsets of genes with different levels of expression, we downloaded the groups defined as 0–20, 20–40, 40–60, 60–80, 80–100 percentile of gene expression from TriTrypDB using transcriptomic data for *T. cruzi* Y strain [49]. Then, using R we associated the predicted TASs to each gene.

## Results

### There is a distinctive average nucleosome arrangement at the TASs in TriTryps

Genome-wide nucleosome mapping by MNase-seq for the parasitic forms present in the insect vector has been performed for TriTryps: *L. major* [35], *T. brucei* [38] and *T. cruzi* [36,37]. Some similarities and some differences from these original works could be inferred. However, a proper systematic comparison is still missing. Therefore, we performed a parallel analysis of the individual datasets following the same informatics workflow for all of them as we described before [37]. The list of raw data we used for every analysis is summarized in detail in S1 Table. To make a fair comparison, it is important to contrast samples that have achieved a similar level of MNase digestion. To corroborate this feature, we represented the length distribution of the sequenced DNA molecules for each sample into histograms (S1 Fig). The levels of digestion achieved for *T. cruzi* and *T. brucei* samples are in a good range for nucleosome core mapping, since most of the sequenced DNA molecules are ~147 bp. In the case of *L. major* the samples are less digested (as observed by the longer main peak in length distribution histograms in both replicates); therefore, we considered this bias for result interpretations. In *T. brucei*, despite the histograms resemble those of *T. cruzi*, it is worth mentioning that the samples were gel-purified before library preparation. Therefore, we cannot rule out that the original samples could have been less digested and that only the nucleosome-size fraction was sequenced.

One of the most interesting features that arose in the original articles is the fact that there is a distinctive average nucleosome arrangement at intergenic regions around the TASs in TriTryps that differs from the rest of the genome. Hence, to make a comparative analysis of average chromatin organization, we predicted the TASs for the three organisms using UTRme [48]. This program makes a prediction of the 5' untranslated region (5'UTR) and as an approximation of the TAS, we used the 5' end of the 5'UTR region predicted with the best score in each case (Fig 1A). A list containing the genomic coordinates for the predicted TASs for each TriTryp is detailed in S2 Table. Consistent with the original works, we observed that average chromatin organization only shows a mild change around the TASs with no regular nucleosome phasing in any TriTryp (Fig 1B and S2A Fig, top panels). Remarkably, we corroborated the presence of a mild NDR around the TASs in *T. cruzi*, a shallower trough preceded by a small footprint of MNase protection in *L. major* and a protection centered at the TAS in *T. brucei,* as previously reported [35–38].

Given that average patterns can mask gene-to-gene variability, we also represented nucleosome occupancy into heatmaps for every individual region relative to the TAS in a 1 kb window. We could observe that the chromatin landscape is not just an average, but it is maintained in most of the detached regions represented for TriTryps (Fig 1B and S2A Fig, lower panels).

To determine the size of the sequenced molecules, that indirectly unveil the size of the molecules responsible for protecting the DNA from MNase digestion, and to know their location relative to the TAS, we represented the data into 2-dimensional plots (2D-plots) as previously described [37,51] (Fig 1C and S2B Fig). Consistently with the length distribution histogram, most of the DNA molecules are ~150 bp for *T. cruzi* and *T. brucei*, in a wider range for the latest, and a bit longer for *L. major*. Note that for *L. major* and *T. brucei* the samples were gel-purified before library preparation and that has implications on the range of the fragments detected in each case. Additionally, 2D-plots exposed that the DNA

molecules protecting the TASs in *T. brucei* or the spliced-out fragment in *L. major* have the size of a nucleosome core particle, consistent with previous reports that describe the presence of a well-positioned nucleosome at those specific points [35,38]. However, given that the original samples were gel-purified, we cannot rule out that additional molecules could be involved in protecting DNA from digestion.

In *T. cruzi*, those DNA protecting molecules around the TASs were not detected, but it is probably due to the extent of the MNase digestion.

Overall, this analysis suggests the presence of some conserved MNase sensitive complex in or near the TASs, although some uniqueness might be involved in each TriTryp.

## An MNase sensitive complex occupies the TASs in *T. brucei*

To explore whether the protection observed at the TASs in *T. brucei* is due to an MNase-sensitive complex sitting in or near that point, we analyzed samples exposed to different levels of digestion using different datasets publicly available (S3 Fig and S1 Table). By representing average nucleosome density relative to the TAS we could observe that, for early digested time points, the TASs are covered by some protecting complex. As the digestion proceeds, the complex is less pronounced, reaching a minimum where a trough is observed (High digestion), consistent with the presence of an MNase sensitive complex (Fig 2A). Moreover, heatmap and 2D-plot representation have enlightened that, at an early digestion point (Low digestion) the complex that protects the TASs is only accessible in part of the genome, and it has a footprint that is heterogeneous in size. This analysis shows how important it is to check the level of digestion reached by a given sample when comparing them, since different regions of the genome are not equally accessible. Therefore, if we compare similar levels of digested sample for *T. brucei* (Fig 2, intermediate digestion) and *T. cruzi* (Fig 1B), we can observe a comparable chromatin organization with NDRs formation at the TASs. This observation suggests a conserved pattern of chromatin landscape with a potential role in mRNA maturation among TriTryps.

## The MNase sensitive complexes protecting the TASs in *T. brucei* and *T. cruzi* are at least partly composed of histones

Despite most of the time DNA protection to MNase digestion is mediated by nucleosomes, on occasions other non-histone binding complexes could be involved [53,54]. To understand the nature of the MNase protection at TASs observed in *T. brucei,* we analyzed data obtained by MNase-ChIP-seq of histone H3 publicly available [33,43] (S1 Table). By analyzing average occupancy of histone H3 relative to the TAS, we could observe that the MNase protection previously observed at the reference point in *T. brucei* disappeared. Instead, we detected a mild trough as reported [33] (Fig 3A, middle panel and S4A Fig, left panel).

As we discussed before, the protection of the TASs is tightly connected to the extent of the sample digestion. To be sure we had not missed any partial histone protection of the TAS due to a differential digestion, we analyzed the average signal of histone H3 in the sequenced molecules not only for those fragments belonging to the nucleosome-size range (120–180 bp, middle panel), but also for dinucleosome-size (180–300 bp, left panel) or subnucleosome-size (50–120 bp, right panel) ranges. We could observe that only when sorting fragments smaller than a nucleosome we could detect a partial protection of the TASs mediated by histones (Fig 3A). Although we cannot prove that the TASs are entirely protected by histones in *T. brucei*, we show that the MNase-sensitive complex sitting at the TASs is at least partly composed of histones. This result was true when using either native or crosslinked chromatin (Fig 3A and S4, respectively).

To investigate if the TASs protection only occurs in *T. brucei* or it could be extended to other trypanosomatids, we performed the same analysis using MNase ChIP-seq data for histone H3 from *T. cruzi* [42], using datasets with comparable levels of MNase digestion (S4B Fig). We could observe that the same chromatin organization was displayed around the TASs in *T. cruzi,* not only in the average representation but also in every region relative to the TAS, as shown in the heatmaps (Fig 3B and S4A Fig, right panel). This observation suggests that chromatin protection and sensitivity to MNase digestion at the TASs are similar for both parasites.

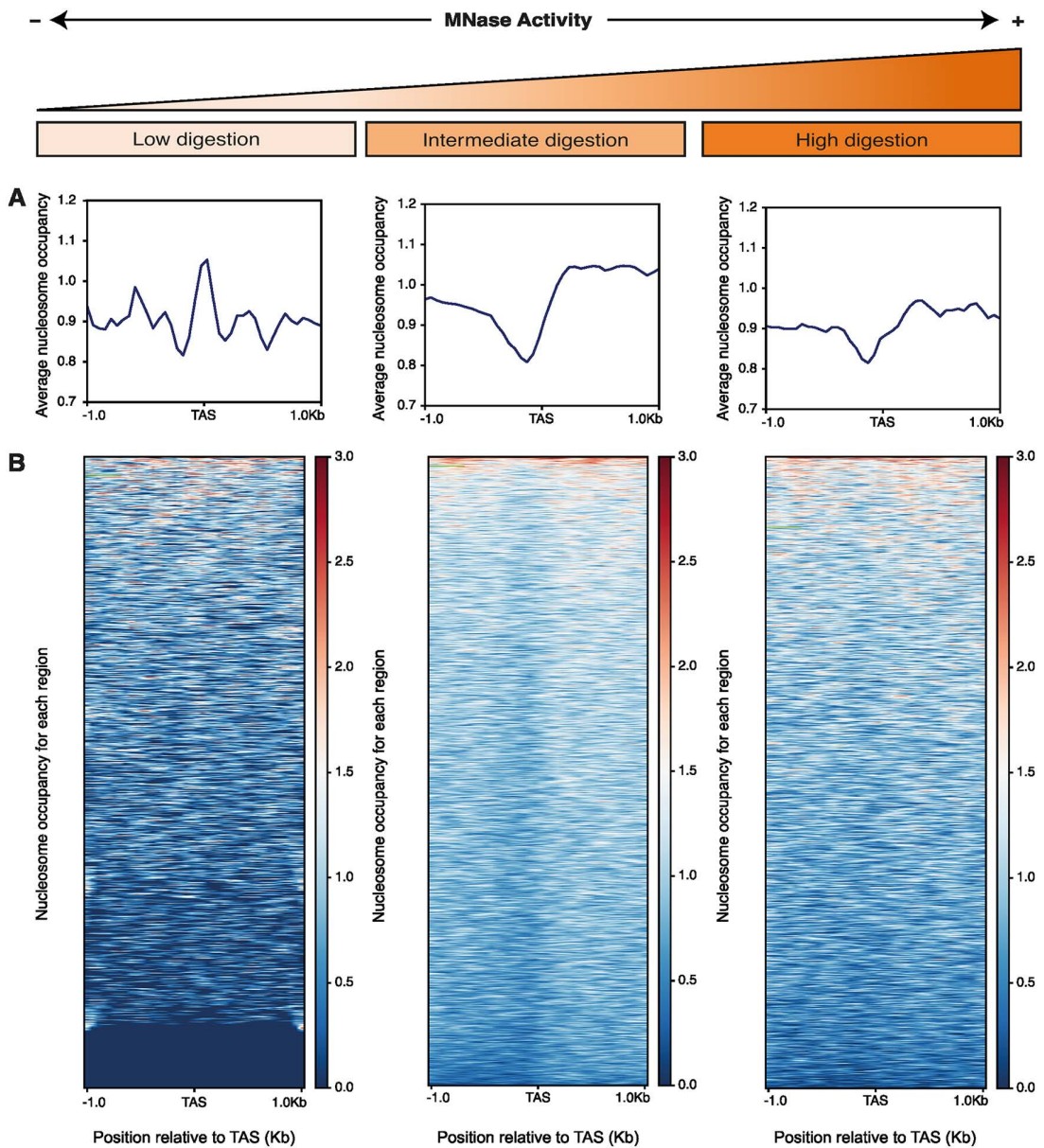

**Fig 2. Differential sensitivity to MNase at the TASs in *T. brucei*.** (A) Average nucleosome occupancy and (B) heatmaps showing nucleosome density relative to the TAS for each region in a 1 kb window for procyclic forms of *T. brucei* exposed to different levels of MNase digestion. Red: High nucleosome density; blue: low nucleosome density. The signals scored for every DNA molecule sequenced (0-500 bp) are represented. Red: High nucleosome density; blue: low nucleosome density. The data sets used in this figure are: Low digestion (GSM5024927), intermediate digestion (GSM5024915) and high digestion (GSM5024921).

## TAS protection involves non-histone components

Despite most of the time DNA protection to MNase digestion is mediated by nucleosomes, on occasions other non-histone binding complexes could be involved [53,54]. Since we observed that only when looking at sub-nucleosome size of a properly digested sample, we were able to detect histone H3, we wonder whether a non-histone component could also be involved.

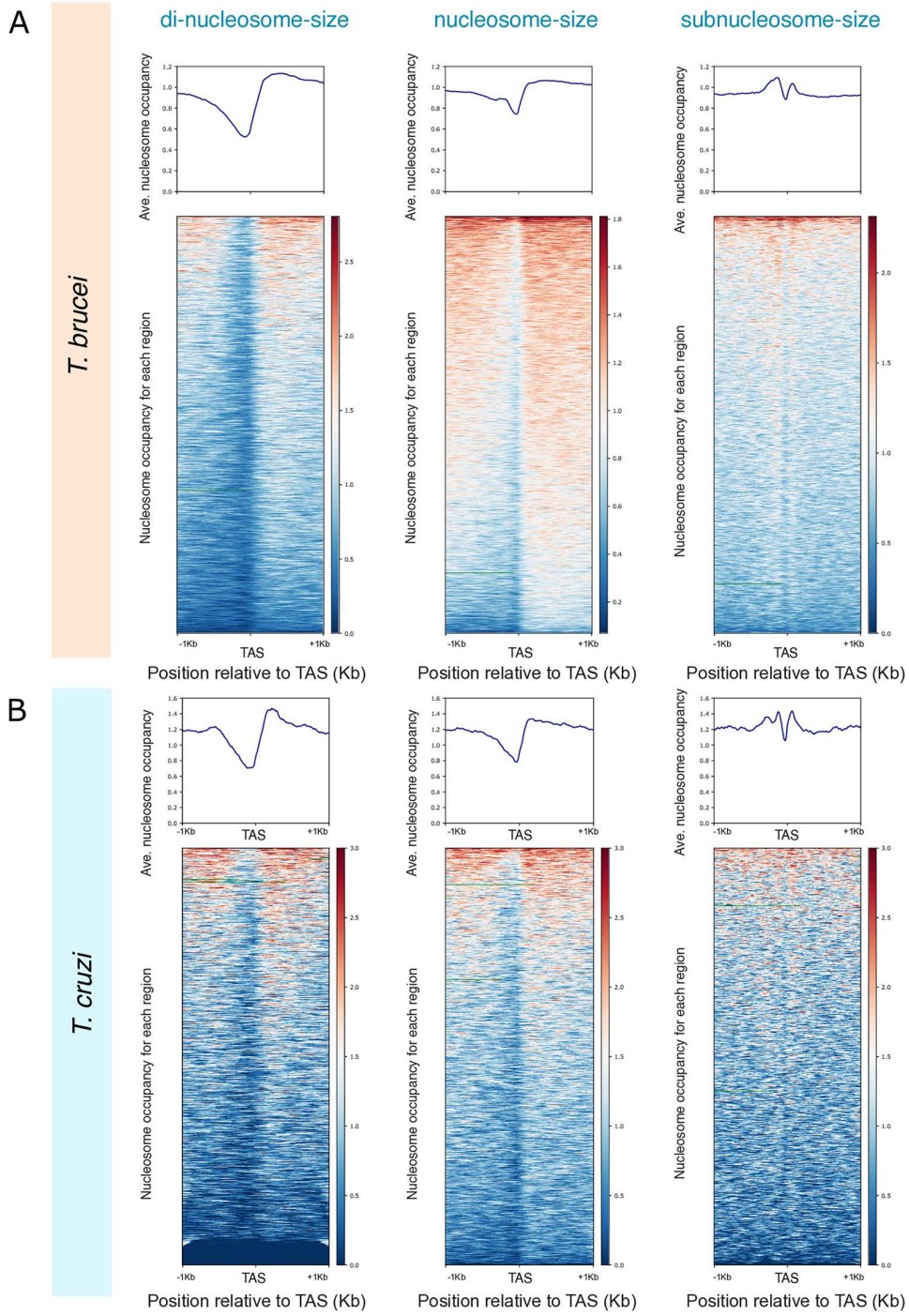

**Fig 3. TASs resistance to MNase digestion is partly mediated by a histone component.** Average H3 occupancy (top panels) and heatmaps (bottom panels) relative to the TAS for one representative experiment of MNase-ChIP-seq of histone H3 (A) *T. brucei* (GSM2586510) and (B) *T. cruzi* (SRR14691958). Red: High nucleosome density; blue: low nucleosome density.

As mentioned above, there are some precedents of MNase protection by tightly bound non-histone proteins at NDR regions in other systems. Given that only a few DNA binding proteins exist in TriTryps, it is hard to postulate a possible candidate for this role in Trypanosomes. Nevertheless, it was previously shown that this mild nucleosome depletion detected at the TAS in *T. brucei* (Fig 3A) co-localize with the accumulation of DNA:RNA hybrids (R-loops), as observed in average occupancy plots detected by DRIP-seq [55]. Here, we corroborated this observation. Furthermore, we could discern that this disposition is not only an average but is conserved at every region around TAS (Fig 4A). Moreover, by representing the H3-IP signal into heatmaps keeping the same sorting used for the R-loops, we could observe that nucleosome depleted regions reflect almost the same genomic regions occupied by R-loops (Fig 4B, bottom panel). Consistently with previous reports, recruitment of RNA polymerase II (Pol II) mimics nucleosome organization around TAS 10 [33]. Additionally, by heatmap representation we could unveil that, as it happens with histone H3, Pol II distribution around each TAS is complementary to the R-loop footprint (Fig 4C). In addition to this, there is a mild correlation between the intensity of the

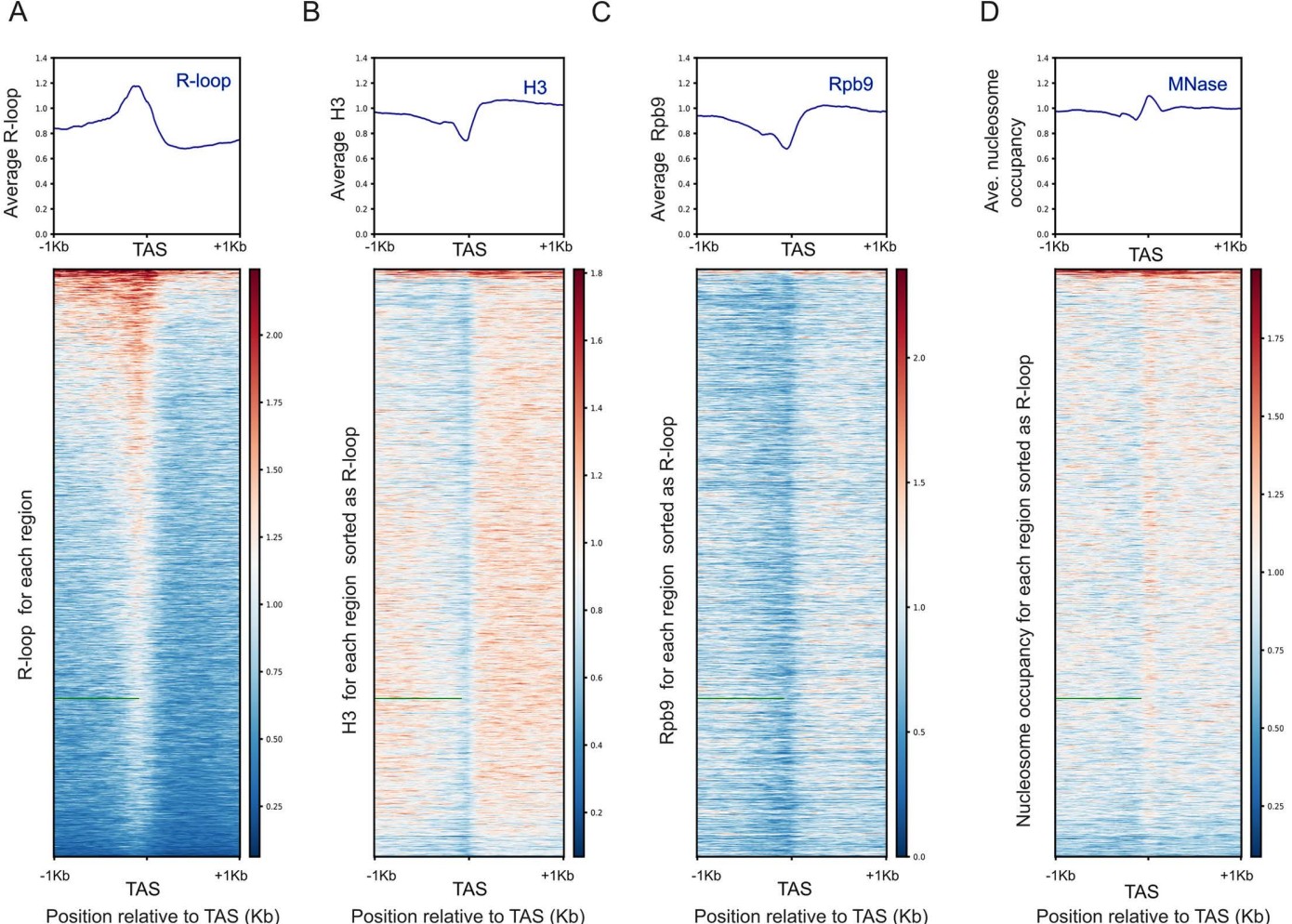

**Fig 4. TAS resistance to MNase digestion includes non-histone components.** Average occupancy (top panels) and heatmaps (bottom panels) relative to TAS for (A) Average H3 signal obtained from H3 MNase-ChIP-seq for *T. brucei* (GSM2586510); (B) R-loops obtained from DRIP-seq (ERR2814820); (C) Rpb9 ChIP-seq (SRR5466331) and (D) MNase-seq (GSM2407366). The regions represented into every heatmap keep the sorting following the distribution of the R-loop signal from higher (Red) to lower (Blue) density.

R-loop signal and the TAS protection from MNase as observed when representing the MNase-seq signals into heatmaps (Fig 4D). Moreover, a similar correspondence between R-loop detection and MNase protection is observed in *L. major* (S5 Fig) [56].

These analysis show that the MNase protection detected at the TAS of early digested samples contain R-loops and might involve R-loop interacting proteins.

### The TASs of single and multi-copy genes are differentially protected by nucleosomes

It was previously described that the genomes of trypanosomatids are compartmentalized into core regions holding mainly single-copy genes and species-specific disruptive regions that encode multigene families [27]. Particularly in *T. cruzi*, these two subsets of genes differ not only in their genomic distribution, but they also present different chromatin organization and gene expression levels. On one hand, single-copy genes display a more open chromatin, higher levels of gene expression and faster transcription rates compared to multi-copy genes [57–59]. To unveil if there was any difference at their TASs, we analyzed their average chromatin organization and we observed that the TASs are differentially protected from MNase digestion. Remarkably, single-copy genes harbor more accessible TASs with a mild NDR, while multi-copy genes show TASs fully occupied by nucleosomes (Fig 5A and S6A Fig, left panel). These chromatin arrangements are consistent with the higher levels of gene expression observed for single-copy genes compared to multi-copy genes (S6A Fig, right panel). This observation suggests that, despite transcription being mainly regulated post-transcriptionally in trypanosomes, chromatin organization might represent another layer of modulation to guarantee the appropriate maturation of the transcripts. To strengthen the functional link between NDR depth and gene maturation, we represented the average chromatin organization relative to the TASs for subsets of genes according to RNA-seq levels (S6B Fig). We observe that the more express the gene the deeper the NDR observed at the TASs. Additionally, by representing FAIRE-seq signals relative to the TAS for these two subsets of genes, we corroborated that the TASs of single-copy genes display a more open chromatin than multi-copy genes (Fig 5B).

Given that DNA sequence is among the major determinants for nucleosome positioning [17] and that it was previously described that in *T. cruzi* the genome compartments present different GC content [27], we wonder whether the difference in nucleosome organization between single and multi-copy genes can be influenced by their DNA sequence. Hence, we analyzed the average frequency of AT and CG containing dinucleotides relative to the TAS for these two groups of genes in a 100 bp window. We observed that while for single-copy genes DNA is enriched in AT containing dinucleotides particularly upstream of the TASs, for multi-copy genes there is some oscillation between AT and GC containing dinucleotides (Fig 5C). Thus, while the long stretches of AT dinucleotides observed in single-copy genes are less likely to assemble into nucleosomes, stretches with a periodic alternation between AT and CG dinucleotides present in multi-copy genes might be more favorable for wrapping around the histone core. We could speculate that the different composition possibly implies that the appropriate transcript maturation might be additionally granted by the underlying DNA sequence in a stage-independent manner. However, future studies in this direction will be required for an in-depth understanding.

## Discussion

Despite gene expression in trypanosomatids is mainly regulated post-transcriptionally, it was shown that the genome of the trypanosomes is organized into chromatin-folding domains underlying that chromatin and DNA accessibility to some extent control gene expression [13,60].

Here, by analyzing MNase-seq data from the parasitic forms present in the insect vector, we made a contrasting study of genome-wide chromatin organization in TriTryps. To do so, we performed a thorough and systematic analysis of the available datasets following the same informatics pipeline for one representative strain of each organism: *T. cruzi* CL Brener, *T. brucei* 427 and *L. major* Friedlin. In contrast with the regular spaced and phased nucleosomes observed in yeast [29], and consistent with previous observations for trypanosomes, they have in common a poorly organized

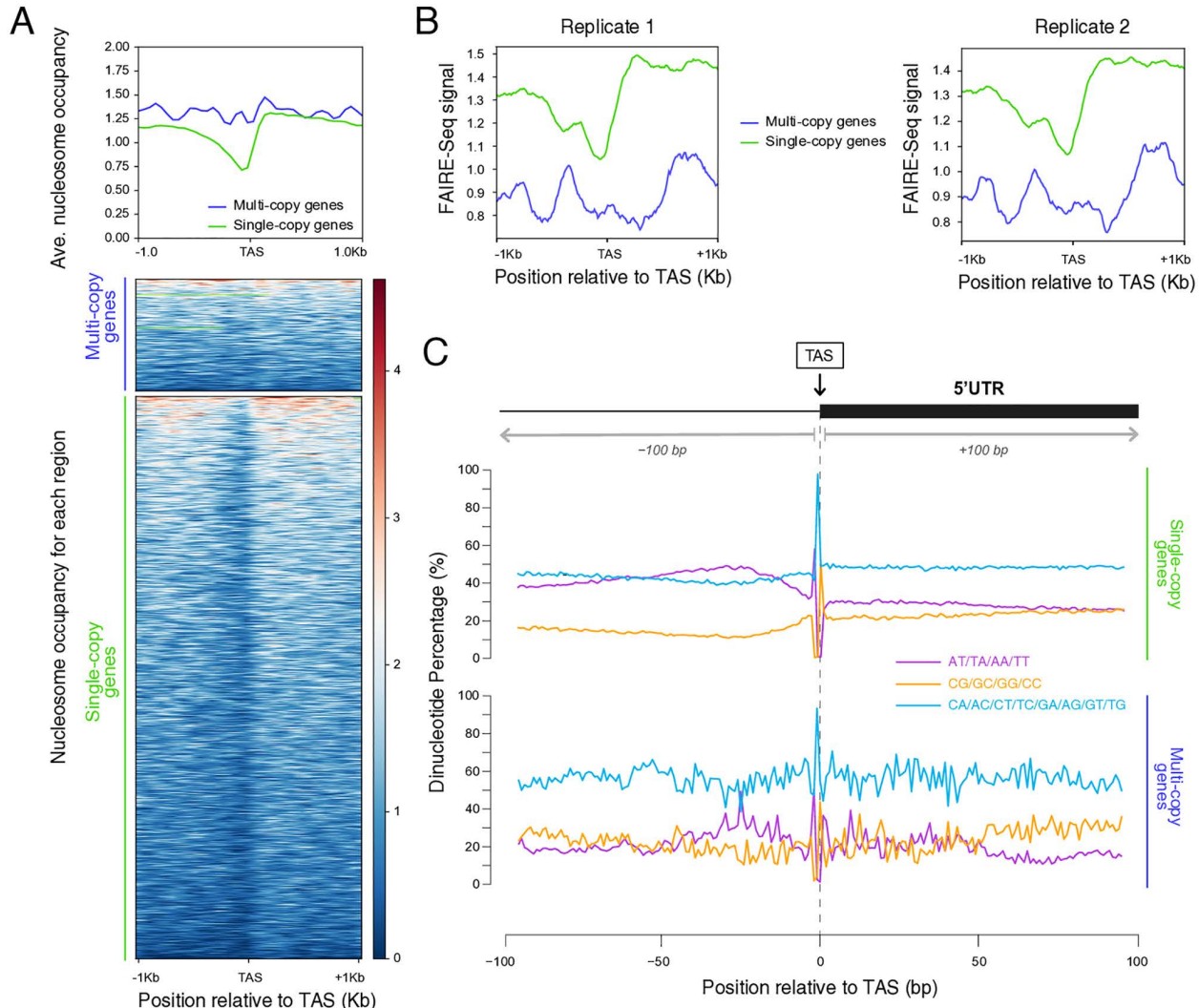

**Fig 5. Chromatin organization at TASs in single and multi-copy gene families.** (A) Average histone H3 occupancy (top panel) and heatmaps (bottom panels) in a 1 kb window relative to the TAS for single (green) and multi-copy genes (blue) using *T. cruzi* histone H3 IP (SRR14691958). Red: High histone H3 density; blue: low histone H3 density. (B) Average FAIRE-seq signals were represented relative to the TAS in a 1 kb window for single (green) and multi-copy genes (blue) of *T. cruzi* using public data (SRR15902298) (left) and (SRR15902297) (right), respectively. (C) Average dinucleotide frequency for AA/TT/AT/TA (purple), CG/CC/GC/CG (orange) and other possible combinations (light blue) in a 100 bp window relative to the TAS for single and multi-copy genes.

chromatin with nucleosomes that are not strikingly positioned or phased, being the most remarkable characteristic the presence of a peculiar change in nucleosome arrangements around the TASs (Fig 1B and S2A Fig).

The earliest genome-wide nucleosome mapping by MNase-seq performed in TriTryps was done in *L. major*, where the presence of a well-positioned nucleosome at the spliced-out region followed by a shallowed trough at the TASs was reported [35]. Afterwards, in *T. brucei*, a similar study revealed a mild nucleosome depletion upstream of the first gene of the DGCs, coincident with divergent strand switch regions but described the presence of a well-positioned nucleosome at the TASs for internal genes [38]. Almost in parallel, MNase-ChIP-seq for histone H3 was performed in *T. brucei* showing a nucleosome depletion upstream of every gene of the DGCs[33]. Later on in *T. cruzi*, a nucleosome depletion

was observed upstream of every gene [36]; and our group described that these NDRs co-localized with the predicted TASs [37].

In this work, we corroborated the original observations and brought to light that the most distinctive feature of average chromatin patterns shared by TriTryps is a peculiar change in average chromatin organization around TASs. Moreover, the change in the chromatin landscape observed at that point is not only an average but is observed for every TAS along their genomes (Fig 1B and S2A Fig). When comparing replicate experiments, some minor differences are noticeable in samples from *T. brucei* and *L. major*. However, given that in both cases samples were gel purified, we cannot over conclude about the impact of different extension in chromatin digestion or the bias introduced by gel purification of the DNA molecules.

By analyzing the extent of sample digestion, in *L. major* and *T. brucei* we exposed that these samples were less digested than those for *T. cruzi*, suggesting that the MNase protecting complex at the spliced-out region or the TASs could have been destroyed during the more extensive digestion experienced by *T. cruzi* samples (Fig 1C, S1 Fig and S2B Fig). Despite the sizes of the sequenced DNA molecules that look similar between *T. cruzi* and *T. brucei*, those from the latter were gel-purified before library preparation. The reason why the MNase protecting complexes were more preserved in *T. brucei* is possibly because they were less digested. From this analysis we could also observe that while the protection is centered at the TAS in *T. brucei*, it is shifted upstream in *L. major*. Whether this is the result of a methodological difference, or it has a biological meaning will require further investigation. However, it is worth mentioning that the GC content around TASs is different in *L. major* compared to trypanosomes [26], that could contribute to explain this differential arrangement. Furthermore, this shift is not only observed for average nucleosome occupancy, but the footprint is also coincident with R-loop detection by DRIP-seq (Fig 1 and S5 Fig).

As discussed before in different eukaryotic systems, using several digestion timepoints helps to understand more deeply what is coded in the chromatin landscape [61–63]. To expose the MNase-sensitive nature of the TASs protecting complex, we analyzed MNase-seq datasets available for *T. brucei* in which the samples achieved different levels of digestion. We could observe that the TASs protection was indeed closely related to the size-range of the sequenced DNA molecules (Fig 2 and S3 Fig). Moreover, it is interesting to compare the heatmaps and 2D-plot representation for the early digestion point (Low digestion)—were the complex that protects the TASs is only accessible in part of the genome, and it has a footprint that is heterogeneous in size (Fig 2 and S3 Fig)—with the sample shown in Fig 1 where the MNase protection is observed in the whole genome (Fig 1, *T. brucei* heatmap). This analysis unveils that different regions of the genome are not equally accessible, highlighting the relevance of checking the level of digestion reached by a given sample. Moreover, the kinetic of digestion of the MNase-protecting complex is faster than the observed for the neighboring nucleosomes (Fig 2), suggesting that TAS protection might be due to a non-nucleosome complex.

Deciphering the nature of the molecule/s that could be bound at those sites of the genome that are less sensitive to MNase, is one of the most relevant questions and a topic of ongoing research in the chromatin field. In yeast, there are a couple of examples where NDRs colocalize with the presence of non-histone complexes at gene promoters and at tDNA genes transcribed by RNA polymerase III [53,54]. To expose the presence of histones in the MNase-sensitive complex protecting the TASs in TriTryps, we analyzed MNase-ChIP-seq data for histone H3 from *T. brucei* and *T. cruzi* from similar levels of digested samples with an optimal range of digestion for nucleosome mapping as previously described [64]. We observed that, when analyzing nucleosome-size (120–180 bp) DNA molecules or longer fragments (180–300 bp), the TASs of either *T. cruzi* or *T. brucei* are mostly nucleosome-depleted. However, when representing fragments smaller than a nucleosome-size (50–120 bp) some histone protection is unmasked (Fig 3 and S4 Fig). This observation suggests that the MNase sensitive complex sitting at the TASs is at least partly composed of histones. In the case of *T. brucei* this observation was true either for native or cross-linked conditions (Fig 3A and S4A, respectively). Unfortunately, there is no similar data available for *L. major*; hence, whether the MNase protecting complex detected at the spliced-out region in *L. major* contains histones remains an open question.

What contributes to NDRs formation in different organisms is a subject of active investigation, but in general, NDRs represent accessible regions that are typically coincidental with regulatory regions. In model organisms, NDRs are related

to promoters, enhancers, origins of replication and tRNA genes. Regarding how those NDRs are formed and maintained, there are several models that involve the concerned activity of transcription factors, histone variants, chromatin remodeling complexes, the transcription initiation machinery and the potential presence of physical barriers [65,66]. Different ATP-dependent chromatin remodeling complexes work in a coordinated manner to keep the NDRs clear, to help to position the + 1 nucleosome and to organize and space nucleosomes on gene bodies [67–70]. In Trypanosomes, there are only a few DNA binding proteins; therefore, it is hard to think about a possible candidate for this role. Instead, this interaction could be bridged by other molecules, such as R-loops. Consistent with our hypothesis, in *T. brucei* and *L. major,* R-loop enrichment was detected at intergenic regions coincident with lower histone density and, among R-loop interacting proteins, some putative trans-splicing factors have been detected [55,71,72]. In Fig 4 and S5 Fig we illustrate the correlation between R-loop formation and histone depletion, but also the remaining concordance with MNase-protection. This counterintuitive observation could be explained by the concomitant presence of histones with other non-histone components, that could protect from MNase digestion at early points but are digested faster than the neighbor nucleosomes as digestion proceeds.

Based on this observation, we propose that in TriTryps the NDRs are formed at the TASs to guarantee the proper maturation of the transcripts when needed. Whether the NDRs favor the appropriate assembly of the trans-splicing machinery or the other way around, is still an open question.

To test the feasibility of this hypothesis, we analyzed separately two subsets of genes: single-copy genes, which usually encode housekeeping functions for the stage of the parasite present in the insect vector; and multi-copy genes, required for infection and immune system evasion [2]. We uncovered that most of the genes belonging to the single-copy gene subset show mild NDRs at the TASs, while multi-copy genes present TASs normally obstructed by nucleosomes (Fig 5A and S6A Fig, left panel). This observation is consistent with previous reports that multi-copy genes are associated with higher nucleosome occupancy, lower levels of expression and transcription rates, as opposed to single-copy genes which display a more open chromatin, higher expression levels and transcription rates. It was also described that in *T. brucei* and *T. cruzi*, the expression of multi-copy genes is also modulated by special isolation in the nucleoplasm [57–59]. Unfortunately, we could not obtain a full TAS list for multi-copy genes for *T. brucei* and *L. major* due to the characteristic of the transcriptomic data used to make UTR predictions and the repetitive nature of their genomes. Adding to the idea that chromatin organization at TAS might be closely related to transcription activity, the analysis of chromatin accessibility at single and multi-copy genes by FAIRE-seq shows a concordance (Fig 5B). Moreover, examination of subsets of genes with different expression levels shows differential protection of the NDRs (S6B Fig).

Finally, we show that these two subsets of genes differ in the dinucleotide content, where single-copy genes differentiate by having a striking asymmetry in the percentage of AT-containing dinucleotides upstream of the TAS (Fig 5C). The fact that single-copy genes are enriched in poly AT tracks upstream of the TASs—more refractory to bend into nucleosomes—while multi-copy genes bear a more periodical alternance between AT and GC dinucleotides—easier to bend around the histone octamer—suggests that DNA sequence might be acting as a spare security gate. DNA sequence might contribute to keeping the TASs more accessible at single-copy genes that require to be expressed most of the time, while it might facilitate nucleosome formation to prevent the unnecessary expression of multi-copy genes, only meant to fulfill a very specific function. Alternatively, trans-acting factors might be required to generate or keep the NDRs but their interaction with DNA or R-loops at the TAS of single-copy genes might be more efficient.

This differential chromatin organization resembles what is observed in more complex organisms where the presence of NDRs is associated with highly transcribed genes, while silenced genes usually have occluded promoters [31]. Given that in Trypanosomes transcription initiation is almost a constitutive process, we propose that the implications of chromatin are associated to modulate the maturation of the polycistronic transcripts into mature monocistronic units in a co-transcriptional manner. Therefore, the chromatin landscape might be modulating the speed of RNA pol II, forcing a pause near TASs and contributing to guarantee an appropriate maturation of the transcripts as previously observed for cis-splicing [73], and this event occurs more frequently at single-copy genes than at multi-copy genes (Fig 6). Future studies focused on mapping the

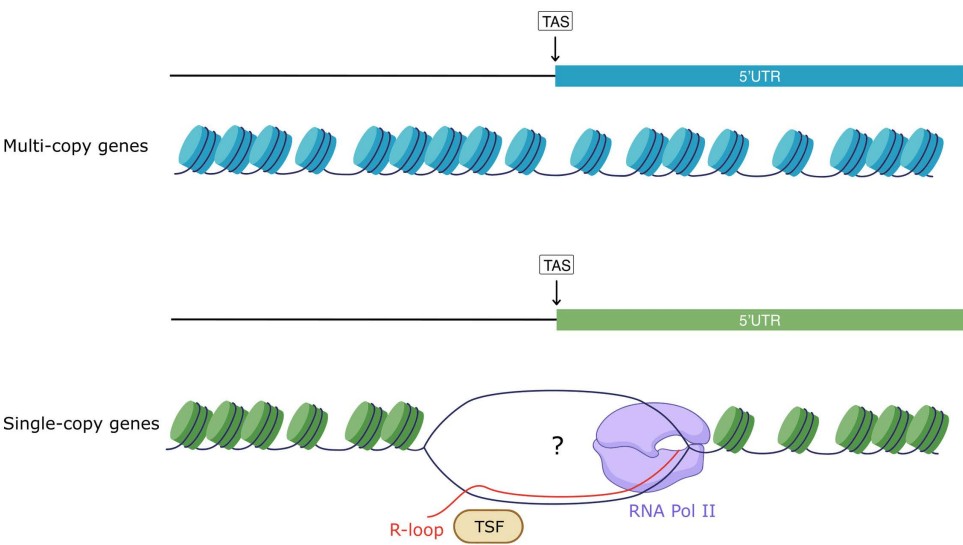

**Fig 6. Differential protection at the TAS at single and multi-copy genes in Trypanosomes.** Schematic representation illustrating the proposed model of chromatin organization at the TAS in TriTryps. Consistent with lower levels of gene expression, multi-copy genes are encoded at genomic regions that are usually less accessible and their TAS are mostly covered by nucleosomes, while single-copy genes, that are transcribed more frequently, display a more open chromatin at the TAS. Moreover, TAS region at single-copy genes harbors R-loops, generated during the co-transcriptional maturation of transcripts, and some trans-splicing factors (TSF) interact through them.

binding of trans-splicing factors along the genomes and studying the dynamics of RNA polymerase II in TriTryps will contribute to answering the unsolved matters.

## Supporting information

**S1 Fig. Length distribution of sequenced DNA.** Length histogram for all nucleosomal DNA sequenced for two replicated experiments for *T. cruzi CL* Brener replicate 1 (GSM5363006) and replicate 2 (GSM5363007) (left panels), *T. brucei* 427 replicate 1 (GSM2407366) and replicate 2 (GSM2407367) (middle panels), and *L. major* Friedlin replicate 1(GSM2179742) and replicate 2 (GSM2179741) (right panels) respectively. Dashed lines indicate the length of the more abundant DNA fragments in the sample.
(TIFF)

**S2 Fig. Average chromatin organization around TASs shows distinctive features in each TriTryp.** (A)Average nucleosome occupancy (top panels), heatmaps for each region in a 1 kb window (bottom panels). The signals scored for DNA molecules in the nucleosomal-size range (120–180 bp) are represented; (B) 2D occupancy plots showing nucleosome density relative to the TAS for all the sequenced DNA for a replicate experiment of *T. cruzi* (GSM5363007), *T. brucei* (GSM2407367) and *L. major* (GSM2179741) respectively. Red: High nucleosome density; blue: low nucleosome density.
(TIFF)

**S3 Fig. TASs protection for differential MNase digestions of chromatin in *T. brucei*.** (A) Length distribution histogram for sequenced DNA molecules for *T. brucei* 427 samples with different extent of MNase digestions. Each data set corresponds to the ones represented in Fig 2. (B) 2D occupancy plots. Red: High nucleosome density; blue: low nucleosome density. Dashed lines indicate the length of the more abundant DNA fragments in the sample. The data sets used in this figure are: Low digestion (GSM5024927), intermediate digestion (GSM5024915) and high digestion (GSM5024921).
(TIF)

**S4 Fig. The TASs of *T. cruzi* and *T. brucei* are mostly depleted of nucleosomes.** (A) Average H3 density (top panels) and heatmaps (bottom panels) for each region in a 1 kb window relative to the TAS. The signals scored for DNA molecules in the nucleosomal-size range (120–180 bp) are represented for *T. brucei* (GSM2586510) and *T. cruzi* (SRR14691957). (B) Length distribution histogram for sequenced DNA molecules for two replicate experiments of MNase-ChIP-seq for H3 for *T. brucei* 427, left panels: top (SRR13477532) and bottom (SRR13477532) and *T. cruzi* CL Brener, right panels: top (SRR14691958) and bottom (SRR14691957).
(TIFF)

**S5 Fig. TAS resistance to MNase digestion might involve non-histone components.** Average occupancy (top panels) and heatmaps (bottom panels) relative to TAS for (A) R-loops obtained from DRIP-seq (ERR12982995); and (B) Average nucleosome occupancy (GSM2179742). In both representations the regions plotted into heatmaps keep the same sorting, following the distribution of the R-loop signal, from higher (Red) to lower (Blue) density.
(TIFF)

**S6 Fig. Differential TASs protection for single and multi-copy gene families in *T. cruzi*.** (A) Average histone H3 occupancy (left panel) and RNA-seq coverage (right panel) for *T. cruzi* CL Brener. (B) Average nucleosome occupancy of *T. cruzi* (GSM5363006) represented for the stratified quintiles according to RNA-seq expression using (SRX574894).
(TIFF)

**S1 Table. Raw data used for every analysis.** Complete information about original data sets and sample accession numbers.
(DOCX)

**S2 Table. Genomic coordinates for predicted TASs for each TriTryp.**
(CSV)

**S3 Table. Genomic coordinates for single-copy and multi-copy genes in *T. cruzi*.**
(CSV)

## Acknowledgments

We are grateful to Dr. David Clark for valuable discussions and to Santiago Carena for reading our manuscript. J.O. and G.D.A are members of the Research Career of CONICET. R.T.Z.S. is Ph.D. fellow supported by ANPCYT and her PhD thesis is carried out at Departamento de Fisiología, Biología Molecular y Celular, Facultad de Ciencias Exactas y Naturales, Universidad de Buenos Aires.

## Author contributions

**Conceptualization:** Guillermo Daniel Alonso, Josefina Ocampo.

**Data curation:** Romina Trinidad Zambrano Siri, Josefina Ocampo.

**Formal analysis:** Romina Trinidad Zambrano Siri, Pablo Smircich, Josefina Ocampo.

**Funding acquisition:** Guillermo Daniel Alonso, Josefina Ocampo.

**Investigation:** Romina Trinidad Zambrano Siri, Paula Beati, Lucas Inchausti, Pablo Smircich, Josefina Ocampo.

**Methodology:** Romina Trinidad Zambrano Siri, Paula Beati, Lucas Inchausti, Pablo Smircich.

**Project administration:** Guillermo Daniel Alonso, Josefina Ocampo.

**Resources:** Guillermo Daniel Alonso.

**Software:** Paula Beati, Pablo Smircich.

**Supervision:** Guillermo Daniel Alonso, Josefina Ocampo.

**Visualization:** Romina Trinidad Zambrano Siri, Josefina Ocampo.

**Writing – original draft:** Josefina Ocampo.

**Writing – review & editing:** Romina Trinidad Zambrano Siri, Paula Beati, Lucas Inchausti, Guillermo Daniel Alonso, Josefina Ocampo.

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
