## [Editor Report · Decision Letter 0]

10 Dec 2025

Dear Dr. Ocampo,

**Going over the documents, we see that the plan you submitted as part of the Review Commons process includes an update of the code deposited at Github (https://github.com/paulati/nucleosome); however, as far as we can tell this has not been done - please provide this information (updated code/pipeline with a link to the repository) so that we can make a decision.**

We look forward to receiving your revised manuscript.

Kind regards,

Alexander F. Palazzo, Ph.D.

Academic Editor

PLOS One

**Journal Requirements:**

1. When submitting your revision, we need you to address these additional requirements. Please ensure that your manuscript meets PLOS ONE's style requirements, including those for file naming. The PLOS ONE style templates can be found at https://journals.plos.org/plosone/s/file?id=wjVg/PLOSOne_formatting_sample_main_body.pdf and https://journals.plos.org/plosone/s/file?id=ba62/PLOSOne_formatting_sample_title_authors_affiliations.pdf 2. We note that the grant information you provided in the ‘Funding Information’ and ‘Financial Disclosure’ sections do not match.  When you resubmit, please ensure that you provide the correct grant numbers for the awards you received for your study in the ‘Funding Information’ section. 3. Thank you for stating the following financial disclosure: International Development Research Centre (IDRC):Guillermo Alonso 109929; Agencia Nacional de Promoción de la Investigación, el Desarrollo Tecnológico y la Innovación (agenciaidiar):Josefina Ocampo PICT-2020-00473; Consejo Nacional de Investigaciones Científicas y Técnicas (CONICET):Josefina Ocampo PIBBA 2020-28720210100100CO; Consejo Nacional de Investigaciones Científicas y Técnicas (CONICET):Guillermo Alonso PIP 2021-2023-03073   Please state what role the funders took in the study.  If the funders had no role, please state: "The funders had no role in study design, data collection and analysis, decision to publish, or preparation of the manuscript." If this statement is not correct you must amend it as needed. Please include this amended Role of Funder statement in your cover letter; we will change the online submission form on your behalf. 4. Thank you for stating the following in the Acknowledgments Section of your manuscript: We are grateful to Dr. David Clark for valuable discussions and to Santiago Carena for reading our manuscript. J.O. and G.D.A are members of the Research Career of CONICET. R.T.Z.S. is Ph.D. fellow supported by ANPCYT and her PhD thesis is carried out at Departamento de Fisiología, Biología Molecular y Celular, Facultad de Ciencias Exactas y Naturales, Universidad de Buenos Aires. We note that you have provided funding information that is not currently declared in your Funding Statement. However, funding information should not appear in the Acknowledgments section or other areas of your manuscript. We will only publish funding information present in the Funding Statement section of the online submission form. Please remove any funding-related text from the manuscript and let us know how you would like to update your Funding Statement. Currently, your Funding Statement reads as follows: International Development Research Centre (IDRC):Guillermo Alonso 109929; Agencia Nacional de Promoción de la Investigación, el Desarrollo Tecnológico y la Innovación (agenciaidiar):Josefina Ocampo PICT-2020-00473; Consejo Nacional de Investigaciones Científicas y Técnicas (CONICET):Josefina Ocampo PIBBA 2020-28720210100100CO; Consejo Nacional de Investigaciones Científicas y Técnicas (CONICET):Guillermo Alonso PIP 2021-2023-03073  Please include your amended statements within your cover letter; we will change the online submission form on your behalf. 5. When completing the data availability statement of the submission form, you indicated that you will make your data available on acceptance. We strongly recommend all authors decide on a data sharing plan before acceptance, as the process can be lengthy and hold up publication timelines. Please note that, though access restrictions are acceptable now, your entire data will need to be made freely accessible if your manuscript is accepted for publication. This policy applies to all data except where public deposition would breach compliance with the protocol approved by your research ethics board. If you are unable to adhere to our open data policy, please kindly revise your statement to explain your reasoning and we will seek the editor's input on an exemption. Please be assured that, once you have provided your new statement, the assessment of your exemption will not hold up the peer review process. 6. Please upload a new copy of Figure 3 as the detail is not clear. Please follow the link for more information:  https://journals.plos.org/plosone/s/figures 7. Please remove your figures from within your manuscript file, leaving only the individual TIFF/EPS image files, uploaded separately. These will be automatically included in the reviewers’ PDF. 8. Please include captions for your Supporting Information files at the end of your manuscript, and update any in-text citations to match accordingly. Please see our Supporting Information guidelines for more information: http://journals.plos.org/plosone/s/supporting-information. 9. If the reviewer comments include a recommendation to cite specific previously published works, please review and evaluate these publications to determine whether they are relevant and should be cited. There is no requirement to cite these works unless the editor has indicated otherwise. 

---

## [Author Response · Author response to Decision Letter 1]

4 Feb 2026

We have checked thoroughly the format requirements and we are uploading a revised version of the manuscript with track changes (Revised Manuscript with Track Changes) and a plain version (Manuscript).

We corroborated that the correct grant numbers for the awards we used for this study are correctly state in the ‘Funding Information’ section.

International Development Research Centre (IDRC):Guillermo Alonso 109929; Agencia Nacional de Promoción de la Investigación, el Desarrollo Tecnológico y la Innovación (agenciaidiar):Josefina Ocampo PICT-2020-00473; Consejo Nacional de Investigaciones Científicas y Técnicas (CONICET):Josefina Ocampo PIBBA 2020-28720210100100CO; Consejo Nacional de Investigaciones Científicas y Técnicas (CONICET):Guillermo Alonso PIP 2021-2023-03073

The funders of our study had no involvement during the execution. Therefore, we state: "The funders had no role in study design, data collection and analysis, decision to publish, or preparation of the manuscript."

We are grateful to Dr. David Clark for valuable discussions and to Santiago Carena for reading our manuscript. J.O. and G.D.A are members of the Research Career of CONICET. R.T.Z.S. is Ph.D. fellow supported by ANPCYT and her PhD thesis is carried out at Departamento de Fisiología, Biología Molecular y Celular, Facultad de Ciencias Exactas y Naturales, Universidad de Buenos Aires.

International Development Research Centre (IDRC):Guillermo Alonso 109929; Agencia Nacional de Promoción de la Investigación, el Desarrollo Tecnológico y la Innovación (agenciaidiar):Josefina Ocampo PICT-2020-00473; Consejo Nacional de Investigaciones Científicas y Técnicas (CONICET):Josefina Ocampo PIBBA 2020-28720210100100CO; Consejo Nacional de Investigaciones Científicas y Técnicas (CONICET):Guillermo Alonso PIP 2021-2023-03073

We have removed the funding-related text from the manuscript. You need to fix a small detail in the current statement bolded below:

International Development Research Centre (IDRC):Guillermo Alonso 109929; Agencia Nacional de Promoción de la Investigación, el Desarrollo Tecnológico y la Innovación (agenciaidiar):Josefina Ocampo PICT-2020-00473; Consejo Nacional de Investigaciones Científicas y Técnicas (CONICET):Josefina Ocampo PIBBA 2020-28720210100100CO; Consejo Nacional de Investigaciones Científicas y Técnicas (CONICET):Guillermo Alonso PIP 2021-2023-03073; Consejo Nacional de Investigaciones Científicas y Técnicas (CONICET).

We are also including the amended statement in the cover letter.

All data used in this manuscript is fully available we have now included a source code for the analytical procedures at https://github.com/romizambrano/TAS.

We have modifyied our statement to make it clearer as follows:

“All relevant data are within this manuscript and its Supporting Information files or extracted from previous published articles. Additionally, a source code for the analytical proceedures here described is available at https://github.com/romizambrano/TAS”.

6. Please upload a new copy of Figure 3 as the detail is not clear. Please follow the link for more information: https://journals.plos.org/plosone/s/figures

We are uploading a new copy of Fig 3 with improved quality.

7. Please remove your figures from within your manuscript file, leaving only the individual TIFF/EPS image files, uploaded separately. These will be automatically included in the reviewers’ PDF.

We have removed the figures from the updated version of the manuscript.

We have included captions for our supporting Information files at the end of the manuscript.

We only updated the reference style according to the Journal style.

No further changes in the reference list have been made.

---

## [Editor Report · Decision Letter 1]

6 Feb 2026

Beyond the transcript: chromatin implications in trans-splicing in Trypanosomatids

PONE-D-25-64632R1

Dear Dr. Ocampo,

We’re pleased to inform you that your manuscript has been judged scientifically suitable for publication and will be formally accepted for publication once it meets all outstanding technical requirements.

Kind regards,

Alexander F. Palazzo, Ph.D.

Academic Editor

PLOS One
---

## [Editor Report · Acceptance letter]

PONE-D-25-64632R1

PLOS One

Dear Dr. Ocampo,

I'm pleased to inform you that your manuscript has been deemed suitable for publication in PLOS One. Congratulations! Your manuscript is now being handed over to our production team.

Kind regards,

on behalf of

Dr. Alexander F. Palazzo

Academic Editor

PLOS One